# AncesTree: An interactive immunoglobulin lineage tree visualizer

**Mathilde Foglierini**[1,2]*, **Leontios Pappas**[1], **Antonio Lanzavecchia**[1], **Davide Corti**[3], **Laurent Perez**[1]

1 Università della Svizzera italiana, Faculty of Biomedical Sciences, Institute for Research in Biomedicine, Bellinzona, Switzerland, 2 Swiss Institute of Bioinformatics, Lausanne, Switzerland, 3 Humabs Biomed SA, Vir Biotechnology, Bellinzona, Switzerland

* mathilde.perez@irb.usi.ch

**Data Availability Statement:** Software, documentation, source code and examples are available at https://github.com/MathildeFogPerez/ancestree

## Abstract

High-throughput sequencing of human immunoglobulin genes allows analysis of antibody repertoires and the reconstruction of clonal lineage evolution. The study of antibodies (Abs) affinity maturation is of specific interest to understand the generation of Abs with high affinity or broadly neutralizing activities. Moreover, phylogenic analysis enables the identification of the key somatic mutations required to achieve optimal antigen binding. The Immcantation framework provides a start-to-finish set of analytical methods for high-throughput adaptive immune receptor repertoire sequencing (AIRR-Seq; Rep-Seq) data. Furthermore, Immcantation's Change-O package has developed IgPhyML, an algorithm designed to build specifically immunoglobulin (Ig) phylogenic trees. Meanwhile Phylip, an algorithm that has been originally developed for applications in ecology and macroevolution, can also be used for the phylogenic reconstruction of antibodies maturation pathway. To complement Ig lineages made by IgPhyML or Dnaml (Phylip), we developed AncesTree, a graphic user interface (GUI) that aims to give researchers the opportunity to interactively explore antibodies clonal evolution. AncesTree displays interactive immunoglobulins phylogenic tree, Ig related mutations and sequence alignments using additional information coming from specialized antibody tools. The GUI is a Java standalone application allowing interaction with Ig tree that can run under Windows, Linux and Mac OS.

This is a *PLOS Computational Biology* Software paper.

## Introduction

Development of Next Generation Sequencing (NGS) methodology and its use for high-throughput sequencing of the Adaptive Immune Receptor Repertoire (AIRR-seq) has provided unprecedented molecular insight into the complexity of the humoral adaptive immune response by generating Ig data sets of 100 million to billions of reads. Different computational methods have been developed to exploit and analyze these data [1]. Retracing the antigen-driven evolution of Ig repertoires by inferring antibody evolution lineages is a powerful

**Funding:** The author(s) received no specific funding for this work.

**Competing interests:** The authors have declared that no competing interests exist.

method to understand how vaccines or pathogens shape the humoral immune response [2–5]. Indeed, Abs maturation is the result of clonal selection during B cell expansion and a clonal lineage is defined as immunoglobulin sequences originating from the same recombination event occurring between the V, D and J segments [6]. B cell receptor (BCR) engagement by a given antigen will trigger somatic hypermutations (SHMs) events generating a large BCR diversity. This process leads to antibodies with mutated Ig variable regions, thus forming a specific B-cell lineage that extends from the naive unmutated B-cells, to somatically hypermutated and class switched memory B or plasma-cells [7]. Lineage tree building requires a common preprocessing step, the clonal lineage assignment [8]. A common starting approach is to initially cluster sequences by their V and J genes and by their CDR3 length. Commonly used tools capable of aligning Ig sequences are MiXCR, IMGT, IgBlast, SONAR, IGoR and iHM-Munealign [9–13]. One of the major drawbacks of the previously mentioned tools is the reliance of the initial alignment with the germline and the exclusion of insertion/deletions (indels) events in the lineage. To circumvent those problems other methods were developed: i) Partis and SONAR [14, 15] can perform both unseeded and seeded lineage assignment ii) Clonify, using a hierarchical clustering approach, performs unseeded lineage assignment [16]. Nonetheless, there is no consensus as to which phylogenetic method is optimal to infer the ancestral evolutionary relationships among Ig sequences [17, 18]. Literally, several methods have been used, such as Levenshtein distance (LD), neighbor joining (NJ), maximum parsimony (MP), maximum likelihood (ML), and Bayesian inference (BEAST) [19–22]. The DNA Maximum Likelihood program (Dnaml) of the PHYLIP package [23], is a ML method that has been originally developed for applications in ecology. It is also commonly used to infer B cell clonal lineages [24–29]. Visualization of the phylogeny is performed using Dendroscope [30, 31]. Meanwhile, a framework was developed to provide a start-to-finish toolbox to process high-throughput AIRR-seq datasets. The Immcantation framework (https://immcantation. readthedocs.io/en/stable/) is currently the gold standard for antibody repertoire analysis. The Change-O tool [32], which is part of Immcantation, was developed to make i) a V(D)J reference alignment standardization after sequences annotation by IMGT/High-VQUEST [33] or IgBlast ii) clonal clustering iii) germline reconstruction iv) conversion and annotation. The IgPhyML algorithm, which is part of Change-O, allows the reconstruction of phylogenic tree by implementing substitution models that correct for the context-sensitive nature of SHM, and combines information from multiple lineages to give more precisely estimated repertoire-wide model parameter estimates. Currently, there is no efficient bioinformatics tool allowing an interactive display of phylogenic tree inferred from Ig sequences. Here we developed AncesTree, an Ig lineage tree visualizer that also integrates information coming from most used antibody bioinformatics tools: IgBlast, IMGT, Change-O, Kabat numbering [34] and BASELINe [35]. AncesTree enables users to interact with a tree containing up to thousands Ig sequences, which were generated by Dnaml or IgPhyML, via the GUI. It is a standalone application that is platform independent and only need JAVA JRE 12 or higher as prerequisite software installed.

## Design and implementation

The AncesTree workflow is presented in **Fig 1**, it consists of three different main steps: Input, Processing and Outputs. Importantly, phylogenetic tree analyses coming from two different tools can be used by AncesTree (Dnaml or Immcantation). If the Dnaml workflow is used, AncesTree will parse the Dnaml output text file. If the Immcantation workflow is used (i.e. RepSeq data), the Change-O tab file in AIRR format, the IgPhyML tab file and it related fasta file (with the reconstructed intermediate sequences) are used as input. Once AncesTree

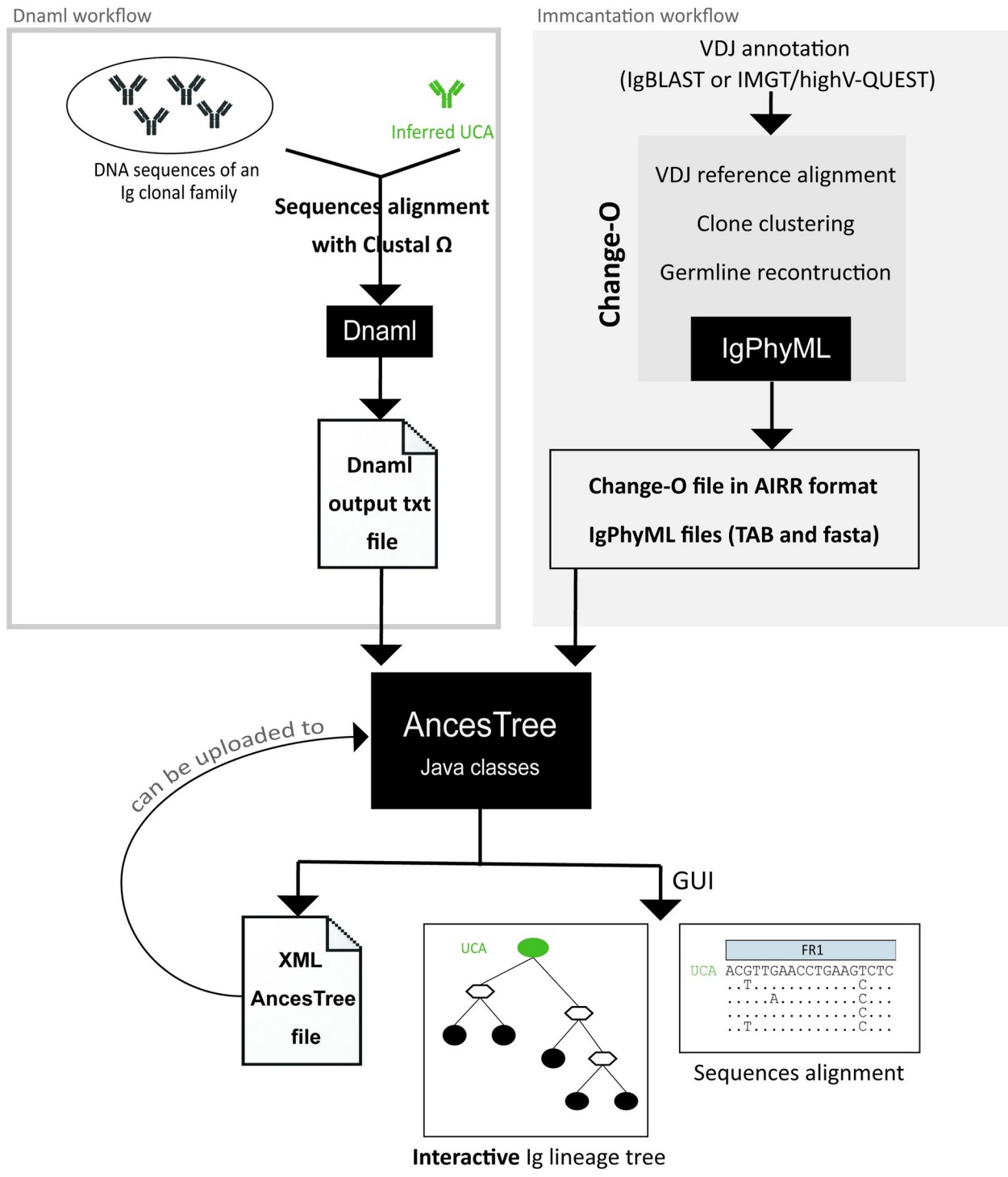

**Fig 1. AncesTree workflow.** Dnaml workflow is used (left part of the figure), DNA sequences are aligned with Clustal Ω and processed by Dnaml, a phylogenetic tree is generated. If the Immcantation workflow is used (right part of the figure), reads are processed through Change-O pipeline and IgPhyML will generate the trees. AncesTree processes the different inputs and reconstructs the phylogenic tree with all information related to Ig. The tree is displayed in a GUI and an Extensible Markup Language (XML) file is produced (that could be used as direct input into AncesTree).

processes the input file(s), it will generate a tree in a graphic interface to allow direct interactivity. Features specific for Ig analysis are included in the GUI.

## Input

AncesTree has two possibilities to display an Ig lineage tree: Dnaml workflow or Immcantation workflow (**Fig 1**). The first one is the Dnaml workflow. The required input for AncesTree usage is the output text file generated by Dnaml. Optionally, a fasta file with data obtained from IMGT can also be used to have full AncesTree features. A clonal family is composed of heavy (or light) V(D)J sequences and their related unmutated common ancestor (UCA). The UCA can be inferred with Antigen Receptor Probabilistic Parser (ARPP) UA Inference software [36] or Cloanalyst [37]. Then, sequences are aligned with Clustal Ω [38] and the generated file in PHYLIP format can be provided for Dnaml. Next, Dnaml is launched with the following settings: 'O' for the outgroup root with the number corresponding to the UCA position provided in the PHYLIP input text file and '5' to reconstruct hypothetical sequences. The generated 'outfile' text file can be used as input for AncesTree. To visualize the different frameworks (FW) and complementary-determining (CDR) regions that composed the Ig variable region, a fasta file can be uploaded. The user provides a fasta file containing the following information: the UCA V(D)J sequence in IMGT format including gaps, and the end positions of each region included in the fasta identifier (separated by a space). This information is easily retrieved using IMGT/V-QUEST [33] with the UCA nucleotide sequence as input. The second possibility for AncesTree to display an Ig lineage tree is through the Immcantation workflow. In most of the cases, this workflow is applied to RepSeq data coming from NGS experiments, but it can also be applied to a small set of sequences. Change-O is processed to run the following steps: i) align to V(D)J reference (sequences are annotated by IMGT/High-VQUEST or IgBlast prior to this step), ii) filter in the productive sequences, cluster sequences by clone, reconstruct germline by clone and iii) convert the change-O file into AIRR format [39]. To infer phylogenic trees with the clones of interest, IgPhyML is run with the '—asr' option, allowing to reconstruct the intermediate sequences of the tree. Of note, this option is only available with the docker image develop version of Immcantation. AncesTree takes as input the Change-O file in AIRR format, the related IgPhyML tab file and the fasta IgPhyML file produced by the '—asr' option. Finally, the user has to specify with a drop down menu which clone id (from the IgPhyML tab file) he wants to be displayed into AncesTree GUI.

## Processing

AncesTree parses the Dnaml output file or the Change-O and IgPhyML files. The Dnaml file is a text file and does not required a tree in Newick format. Naturally, the relationship between the different nodes of the tree is already stored, in addition to the sequence of each node in the Dnaml output text file. Conversely, in the case of IgPhyML input, the tree is reconstructed from a newick format. The theoretical intermediate reconstructed sequences are renamed branch points (BPs). In cases of ambiguous nucleotide notation (IUPAC nomenclature) with the Dnaml input file, AncesTree selects the nucleotide with the highest probability based on the Ig sequences retrieved after this BP. Of note, IgPhyML already makes this correction with the '—asr' option set to 0.1. AncesTree has the ability to collapse a node if the sequences are identical, for example in the case of a theoretical BP corresponding to an existing Ig. Moreover, AncesTree will also draw different nodes clustered together in the case of identical Ig sequences, thus providing a clear topology view of the tree.

## Outputs

After running AncesTree, a sub-folder is automatically created in the 'output' folder using the name of the Dnaml output file or the name of the IgPhyML file with the selected clone id. The

resulting folder will contain all produced files such as a XML file that can be used for direct loading into the GUI.

AncesTree displays the processed tree in the main panel of the GUI (**Fig 2A**). The number of nucleotide and amino acid mutations are written on the edge between each node/sequence (with amino acid mutations shown in parenthesis) and it is clickable, enabling the opening of a new window frame that displays the detailed location of each mutation (**Fig 2B**). Of note, the color of the box around each mutated codon indicates whether the mutation is replacement (R) in red or silent (S) in green. This information is also available as R/S numbers under each region. The user can view the amino acid (a.a.) mutations, and have access by default to the Kabat numbering of the related a.a. position (without internet access, AncesTree will use the a. a. absolute position). To obtain the nucleotide or protein sequence of a node, the user can click on it (**Fig 2C**). The user also has the possibility to enter the $EC_{50}$ for the specified Ig. The sequence alignments (DNA or protein) are accessible in a new frame via the 'Menu' button on

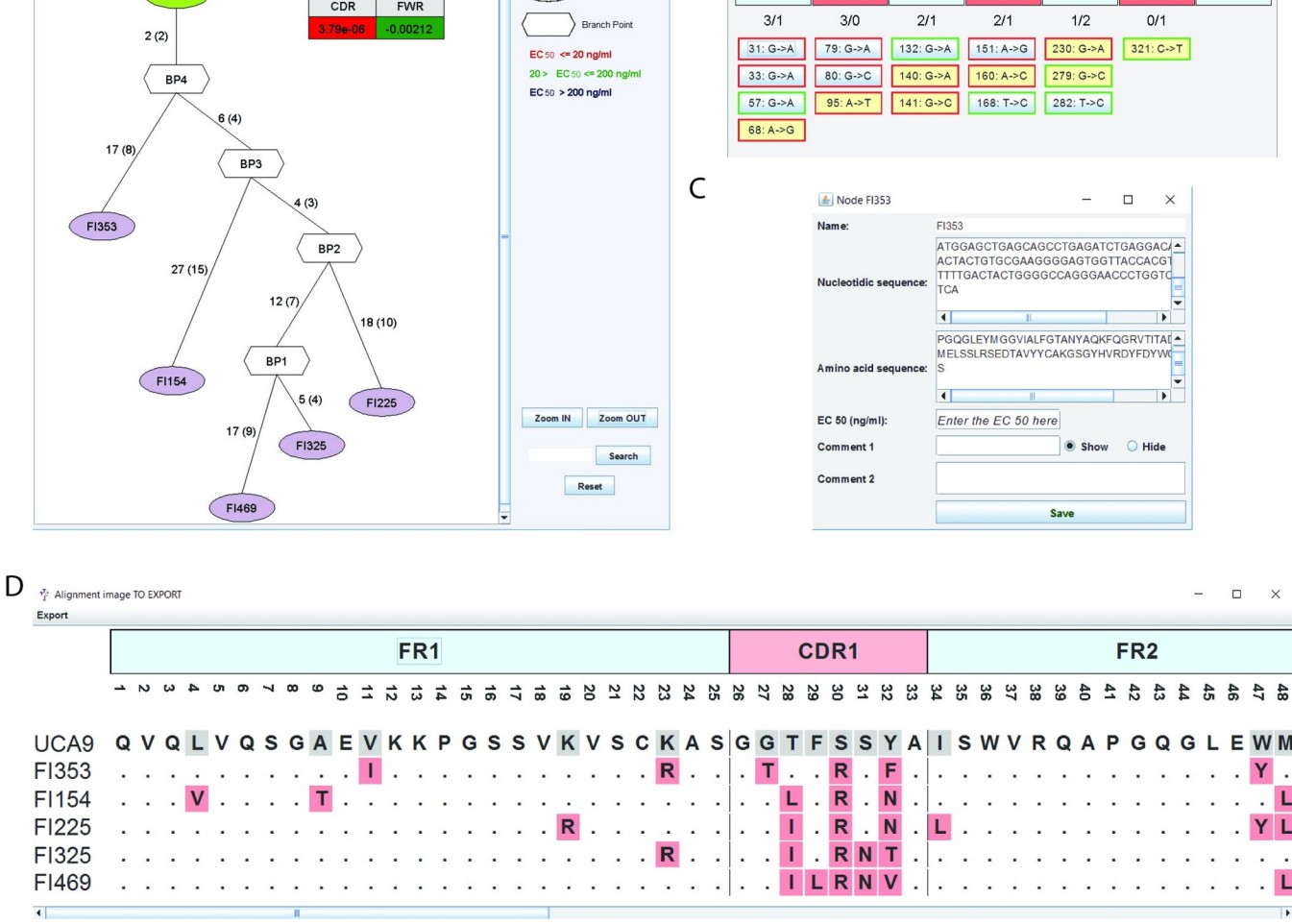

**Fig 2. Snapshot of AncesTree GUI.** (**A**) The tree generated by Dnaml is displayed in the main panel. The BASELINe analysis for the clonal family is displayed in the right upper corner. (**B**) The mutations between two nodes can be displayed in a separate window and they are positioned using IMGT sequence annotation. (**C**) The user can have access to each specific node to obtain the related sequences (DNA or protein) and add comments. (**D**) An alignment is generated with the UCA appearing in the first lane, and a ruler indicates the different regions that compose an Ig sequence.

the top (**Fig 2D**). Finally, the alignment view is customizable: the sequences can be selected or deselected, as well as different positions or regions. Different color modes can be chosen.

If the user is interested in a BASELINe analysis of a clonal family of interest, and if the optional input fasta file (with the UCA VDJ sequence including gaps) was provided with the Dnaml input, AncesTree will automatically generates the fasta input file needed for this software (http://selection.med.yale.edu/baseline/). Once BASELINe has finished to process, the output file can be loaded into AncesTree to have a nice graphic view of antigen-driven selection occurring for this particular clonal family. All graph generated can be exported in PNG or EPS format and the alignment can be exported as Tab-separated Values (TSV) file.

## Results

To demonstrate the utility of AncesTree we analyzed a case study by performing the analysis of an Ig lineage tree targeting the fusion protein (F) of the Respiratory Syncytial Virus (RSV). RSV is an enveloped RNA virus belonging to the recently defined *Pneumoviridae* family [40]. Infection of healthy adults by RSV typically results in mild respiratory symptoms. However, viral infection of infants and older adults, accounts for a substantial hospitalization burden in both age groups [41]. Indeed, RSV infection is the second cause of infant mortality worldwide after malaria [42]. Understanding the immunological basis for the development of potent neutralizing antibodies is a key step for the development of an effective vaccine for RSV.

### Case study: Exploration of Ig lineage targeting the Fusion protein of the Respiratory Syncytial Virus (F-RSV)

To demonstrate the practical use of AncesTree, we re-analyzed an Ig dataset generated post infection by Respiratory Syncytial Viral infection (HRSV). The dataset was collected by isolating antibodies direct against the F-RSV protein, a class I fusion protein mediating viral entry into host cells [43]. The Ig sequences were clustered by grouping antibodies sharing the same VH and VL gene usage, HCDR3 length and identity (at least 85% for HCDR3). Among the clusters generated, we chose Igs targeting the antigenic site V of F-RSV located near amino acid 447 between the α3 helix and β3/β4 hairpin of F-RSV in prefusion (**Fig 3A**). About 70% of the mAbs targeting this site use the same VH and VL germline pair (VH1–18 and VK2–30) [43–45]. We identified an Ig family of interest containing potent neutralizers targeting site V with one outlier, the mAb ADI-14576, being less potent and with a 10-fold decrease in binding affinity (**Fig 3B**). We used Dnaml to generate a VH sequences phylogenic tree and launched AncesTree to analyze and interact with the produced phylogenic tree (**Fig 3C**). The $EC_{50}$ (ng/ml) related to the neutralization assay against RSV subtype A is reported in each node (of note, $EC_{50}$ against subtype B is in the same range for each Ig). Surprisingly, a common mutation 92: G->A (kabat position 31: S ->N) is shared between all the Igs, except for ADI-14576 that does not share this mutation. The alignment of the Ig protein sequences highlights clearly this shared mutation (**Fig 3D**). A result suggesting that ADI-14576 underwent less affinity maturation and therefore diverges from all the other family members. Interestingly, the 31: S->N mutation is located in the HCDR1 and asparagine residues are often involved in protein binding sites. It is tempting to speculate that the Serine to Asparagine substitution is in part responsible for the higher potency and binding titer of the antibodies.

### Concluding remarks

To summarize, we developed an intuitive, easy and interactive GUI allowing the visualization and exploration of antibody clonal evolution. Our application is open access and platform independent. AncesTree only needs the file(s) that can be produced either by Dnaml or by

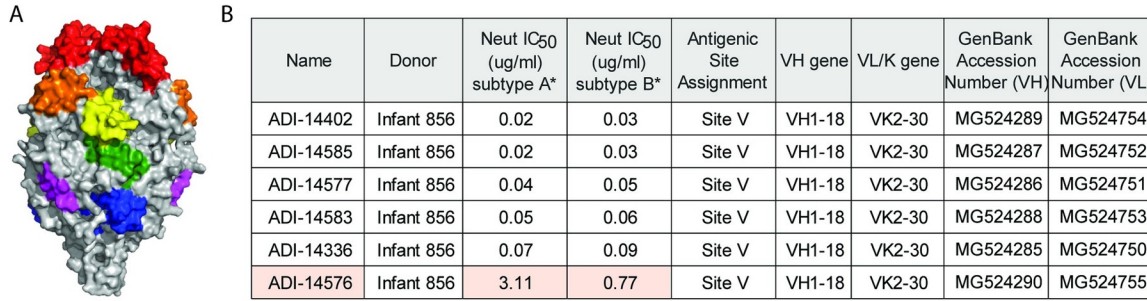

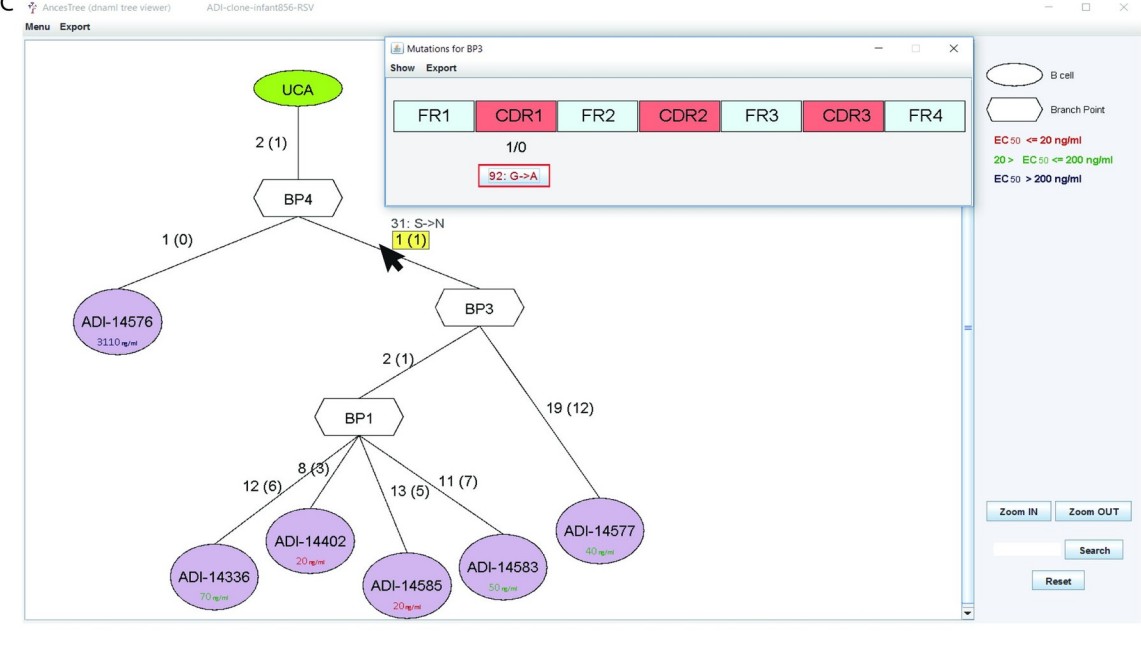

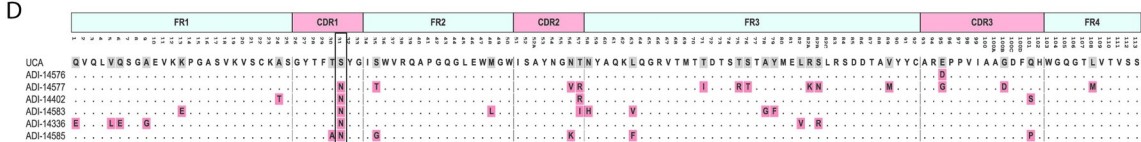

**Fig 3. Clonal family against F-RSV protein antigenic site V. (A)** Shown is the prefusion conformation of F-RSV trimer (PDB ID: 4MMU) [46]. The antigenic sites are colored, site Ø (red), I (blue), II (yellow), III (green), IV (purple) and V (orange). **(B)** Table showing the different characteristic of a mAbs clonal family isolated from an infant (≥ 6 months) after RSV infection. The Igs neutralization titers are shown as well as their related Germline annotations. ADI-14576 is highlighted because of is lower neutralization value in comparison to the other mAbs of the same clonal family. Phylogenetic analysis of the VH chain of a clonal family F-RSV specific. **(C)** Phylogenic tree displayed in AncesTree where the user clicked on the mutation shared by all Igs below BP3 node (31: S->N). **(D)** Protein alignment of the different Ig sequences, the mutation 31: S->N is boxed.

IgPhyML, which is one of the most used tool for antibody repertoire analysis. In the latter, AncesTree processed the Change-O file in AIRR format that allows to share RepSeq data in a standardized format. The possibility to visualize tree independently of the used pipeline allows a broad AncesTree usage. AncesTree was successfully tested with phylogenic trees up to 500 unique sequences processed by IgPhyML and more than a thousand sequences processed by Dnaml. Our interface will provide the users a practical tool containing several useful features that will be of high utility for the immunologists' community and especially those with little or no computational skills.

## Acknowledgments

The authors acknowledge present and past members of the Lanzavecchia's group for comments and feedback on the software.

## Author Contributions

**Conceptualization:** Mathilde Foglierini.

**Data curation:** Mathilde Foglierini.

**Formal analysis:** Mathilde Foglierini, Leontios Pappas.

**Investigation:** Mathilde Foglierini, Laurent Perez.

**Methodology:** Mathilde Foglierini.

**Project administration:** Antonio Lanzavecchia.

**Resources:** Antonio Lanzavecchia.

**Software:** Mathilde Foglierini.

**Supervision:** Antonio Lanzavecchia, Davide Corti, Laurent Perez.

**Validation:** Leontios Pappas.

**Visualization:** Mathilde Foglierini.

**Writing – original draft:** Mathilde Foglierini, Laurent Perez.

**Writing – review & editing:** Mathilde Foglierini, Davide Corti, Laurent Perez.

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
