## [Decision Letter · Decision Letter 0]

25 Apr 2020

Dear Mrs Foglierini,

Thank you very much for submitting your manuscript "AncesTree: an interactive immunoglobulin lineage tree visualizer" for consideration at PLOS Computational Biology.

As with all papers reviewed by the journal, your manuscript was reviewed by members of the editorial board and by several independent reviewers. In light of the reviews (below this email), we would like to invite the resubmission of a significantly-revised version that takes into account the reviewers' comments.

We cannot make any decision about publication until we have seen the revised manuscript and your response to the reviewers' comments. Your revised manuscript is also likely to be sent to reviewers for further evaluation.

Sincerely,

Mihaela Pertea

Software Editor

PLOS Computational Biology

Mihaela Pertea

Software Editor

PLOS Computational Biology

Reviewer's Responses to Questions

**Comments to the Authors:**

Reviewer #1: The authors present a graphical user interface for visualizing B cell clonal lineage trees. The program takes results from Phylip, a computational phylogenetics package for inferring evolutionary trees, as an input and allows the user to visually explore the constructed trees by providing features such as annotation of somatic hypermutations or IC50 on the tree edges and annotation of tree nodes (i.e. Ig sequences) with IMGT and BASELINe tools.

The interface provides a good user experience in terms of available features but lacks broader support of alternative B-cell phylogenetic inference tools.

The program is being distributed as a .jar-package which makes it easy to install and run on the provided tutorial data.

Major issues:

- The authors built their interface upon Phylip, which is indeed a widely used tool for constructing phylogenetic trees. However, adding support of other AIRR data processing packages (such as Immcantation, including Change-O and IgPhyML that also uses Phylip algorithm) would greatly increase the number of potential users that can profit from this great UI. Indeed, the AncesTree should be able to run on any tree independent of how the tree was constructed. Thus, the importance given to phylip, especially in the abstract, is not understandable.

- An issue that can hinder the use of the program is the need for a user to decompose the repertoire into clonal lineages by themselves prior to Ancestree analysis, which may be done by Change-O or Clonify. Why is the lineage decomposition not provided out of the box by Ancestree?

- The link to the wiki https://bitbucket.org/mathildefog/ancestree/wiki/ is closed: a user from the outside would not have access to it.

- The graphical interface has a lot of minor visual bugs. Each of the bugs is not critical on its own, but in total, they spoil the impression that the appearance of the interface leaves (see attachment).

- The authors did not include a quantification of how large datasets may be to be feasibly analysed with AncesTree?

- Figure 3 is not a data figure (or unrelated to the overall purpose of the paper) and should therefore be merged with Figure 4 or moved to Supplementary.

Minor issues:

- The trees in the tutorial video and the example data differ.

- The manuscript text requires language editing (many typos/grammar mistakes)

- Add reference to change-o

Reviewer #2: The authors of "AncesTree: an interactive immunoglobulin lineage tree visualizer" describe a software tool aimed at facilitating the analysis of B-cell receptor (BCR) evolution and exploration of BCR lineage trees. I find this tool to be of high utility for the community, especially immunologists with little or no computational skills. While I find this tool quite versatile, it seems there is an issue with supported input formats.

1) There is a standard coming from AIRR community [https://docs.airr-community.org/en/latest/] that provides a comprehensive description of T- and B-cell receptor repertoire sequencing data. In order to be useful for a wider community I suggest incorporating support for AIRR format in the software.

2) In line with previous point, tools like MIXCR and IgBlast are widely used to analyze large amounts of B-cell receptor sequencing data, while IMGT web server is limited to a certain number of input reads and is unfeasible for large sequencing datasets. Authors should consider implementing a mean to import results produced by MIXCR and IgBlast.

3) There is no comparison with Immcantation suite [https://immcantation.readthedocs.io/en/stable/], nor it is mentioned in discussion. This suite is widely used for B-cell lineage analysis so authors should comment on AncesTree features that are not present in this software.

**Have all data underlying the figures and results presented in the manuscript been provided?**

Reviewer #1: Yes

Reviewer #2: Yes

PLOS authors have the option to publish the peer review history of their article (what does this mean?). If published, this will include your full peer review and any attached files.

Reviewer #1: No

Reviewer #2: No
---

## [Decision Letter · Decision Letter 1]

14 Jun 2020

Dear Mrs Foglierini,

We are pleased to inform you that your manuscript 'AncesTree: an interactive immunoglobulin lineage tree visualizer' has been provisionally accepted for publication in PLOS Computational Biology.

Best regards,

Mihaela Pertea

Software Editor

PLOS Computational Biology

Mihaela Pertea

Software Editor

PLOS Computational Biology

Reviewer's Responses to Questions

**Comments to the Authors:**

Reviewer #1: The authors have addressed all comments – if not already done, it would be great if they could add to the manuscript that there is no theoretical upper limit of sequence input.

Reviewer #2: After careful consideration, I find my request regarding MIXCR to be out of scope of present study: it is up the developer to implement an AIRR-compliable format, implementing IgBlast support is enough to resolve the issue I've raised. I have no further comments.

**Have all data underlying the figures and results presented in the manuscript been provided?**

Reviewer #1: Yes

Reviewer #2: Yes

PLOS authors have the option to publish the peer review history of their article (what does this mean?). If published, this will include your full peer review and any attached files.

Reviewer #1: No

Reviewer #2: No

---

## [Editor Report · Acceptance letter]

1 Jul 2020

PCOMPBIOL-D-20-00234R1 

AncesTree: An interactive immunoglobulin lineage tree visualizer

Dear Dr Foglierini,

I am pleased to inform you that your manuscript has been formally accepted for publication in PLOS Computational Biology. Your manuscript is now with our production department and you will be notified of the publication date in due course.

With kind regards,

Laura Mallard
